# Construction of a prognostic prediction model for concurrent radiotherapy in cervical cancer using GEO and TCGA databases with preliminary validation analysis

Siqi Yang[1], Liting Liu[1], Qiuyue Su[1], Jianan Wang[1], Jingqi Xia[1], Xinyao Zhao[1], Yajuan Sun[2]*, Shanshan Yang[1]*

1 Department of Gynecological Radiotherapy, Harbin Medical University Cancer Hospital, Harbin, China,
2 Department of Imaging Center, Harbin Medical University Cancer Hospital, Harbin, China

* yajuan-sun@163.com (YS); yangshanshan@hrbmu.edu.cn (SY)

## Abstract

### Introduction

Radiotherapy is a primary treatment for intermediate and advanced cervical cancer (CC). Resistance to radiotherapy is a principal reason for treatment failure in synchronous applications, yet the molecular mechanisms remain poorly understood. Identifying reliable prognostic markers to predict and evaluate patient outcomes is essential for advancing therapeutic strategies. This study aims to address this need by developing a prognostic prediction model for concurrent radiotherapy in CC, utilizing both single-cell RNA sequencing (scRNA-seq) and bulk RNA sequencing data.

### Methods

The research began by screening for co-expressed genes using samples from two GEO datasets (GSE236738 and GSE56363). To pinpoint target genes that exhibit significant co-expression, both univariate and multivariate Cox regression analyses were conducted, facilitating the development of prognostic prediction models. The clinical significance of these models was confirmed through the analysis of 144 CC samples sourced from the TCGA database, utilizing Kaplan-Meier survival curves, ROC curve analyses, and Spearman's correlation tests to investigate the relationships between gene expression and the levels of immune cell infiltration. IHC assays were conducted to further validate the prognostic potential of the identified target genes in CC patients.

### Results

Prognostic models for four target genes—MPP5, SNX7, LSM12, and GALNT3—showed significant predictive value for the outcomes of CC patients undergoing

**Data availability statement:** All relevant data are within the manuscript and its Supporting Information files.

**Funding:** This study was supported by National Natural Science Foundation of China (82373207), Climing program of Harbin Medical University Cancer Hospital (PDYS2024-06).

**Competing interests:** The authors have declared that no competing interests exist.

radiotherapy, as demonstrated using the GSE236738 and GSE56363 datasets. The prognostic efficacy of the model was illustrated through scatter plots and calibration curves. Additionally, the model exhibited significant associations with tumor immune infiltration, immune checkpoints, and chemotherapeutic drug sensitivity. Immunohistochemistry (IHC) on clinical tumor samples confirmed that the protein expression levels of MPP5, SNX7, LSM12, and GALNT3 were distinctively predictive for CC patients.

## Conclusion

The results indicate that MPP5, SNX7, LSM12, and GALNT3 are significantly associated with radiotherapy sensitivity in CC cells. A prognostic risk model based on these genes demonstrated strong predictive capabilities for patient outcomes in radiotherapy, suggesting these genes as effective predictors and potential therapeutic targets for treating CC.

## 1 Introduction

CC ranks as the fourth most prevalent malignancy among women globally and stands as the fourth primary contributor to cancer-related mortality within this demographic, with an estimated 604,000 new diagnoses and 342,000 fatalities documented worldwide in the year 2020. With the enhancement of global living standards and the progression of diagnostic and treatment techniques, CC has emerged as the third leading cause of cancer-related deaths in young women [1,2]. According to NCCN guidelines, the treatment for CC typically involves surgical intervention in early stages and concurrent radiotherapy in intermediate to advanced stages [3]. Treatment strategies may incorporate a combination of chemotherapy, radiotherapy, biotherapy, and immunotherapy, depending on the disease stage. Radiotherapy, particularly crucial in the treatment of mid to late-stage CC, aims to minimize local recurrence and enhance patient survival.

Nevertheless, some patients exhibit significant resistance to radiation therapy, which adversely impacts treatment outcomes and prognosis. The phenomenon of radiation resistance in cancer can often be linked to various mechanisms, such as the repair processes of DNA damage caused by radiotherapy [4,5], disruptions in the cell cycle [6,7], the enduring presence of cancer stem cells (CSCs) [8,9], and the occurrence of epithelial-mesenchymal transition (EMT) [10–12]. Comprehending these mechanisms is essential for advancing therapeutic approaches, tailoring treatment regimens, and increasing patient survival outcomes. The tumor microenvironment (TME) [13], comprising stromal cells, signaling molecules, immune cells, and the extracellular matrix (ECM), is crucial in the progression of CC and the development of treatment resistance. This microenvironment is characterized by intricate interactions with tumor cells, contributing to tumor proliferation, migration, and immune evasion.

Hence, identifying biomarkers related to radiotherapy sensitivity is important for predicting treatment effects and tumor progression. While the PAX1 gene [14] has

been proposed as a predictive biomarker for radiation sensitivity in CC, few studies have explored the prognostic value of genes associated with radiotherapy response through the integration of multiple genetic factors and the construction of predictive models. Combining several biomarkers into a unified model could significantly improve prognostic accuracy.

Single-Cell RNA sequencing (scRNA-seq) provides a high-resolution genome-wide analysis of gene expression, offering valuable insights into cellular heterogeneity, transitions between cell states, and intercellular communication within complex tissues. It also facilitates the analysis of frozen tissues, which can be preserved for extended periods, enabling longitudinal patient studies [15,16]. Moreover, scRNA-seq, a dissociation-free method, minimizes gene expression disruption caused by enzymatic digestion required for traditional sequencing approaches [17–19].

In this study, we constructed and confirmed a prognostic risk model by utilizing both GEO dataset and TCGA cohort data, incorporating scRNA-seq, bulk RNA sequencing, and Lasso regression analysis. The chosen genes underwent in vitro testing to evaluate their efficacy in forecasting radiotherapy resistance and tumor advancement. These findings could offer potential predictive biomarkers of radiotherapy response and new therapeutic targets to enhance response rates in patients with locally advanced CC.

## 2 Materials and methods

### 2.1 Patient sample collection

For this study, scRNA-seq data were downloaded from the GEO database, specifically from dataset GSE236738 [20], which includes six tumor tissue samples from three CC patients evaluated before treatment (TN) and after CCRT. Additionally, relevant samples were incorporated from the GEO database by searching keywords "CC," "radiotherapy," and "resistance." Dataset GSE56363 [21] was selected as the primary dataset for analysis, involving 21 patients with locally advanced squamous cell carcinoma [FIGO stage IIB-IIIB]. Twelve patients who achieved complete remission (CR) were categorized into the CR group, while the remaining nine, exhibiting partial remission (PR) or stable disease (SD), were placed in the NCR group. Further, transcriptomic data along with survival information for 144 CC patients were retrieved from the TCGA database. All mRNA expressions from GEO and TCGA data were manually downloaded and processed using R software.

### 2.2 ScRNA-seq

In order to reduce the dimensionality of the GSE236738 dataset, Principal Component Analysis [PCA] and UMAP were applied. Gene expression profiles were used to do cell clustering using the Seurat package in R. Marker genes were defined by criteria of an adjusted p-value < 0.05 and |log2[fold change]| > 0.25.

### 2.3 Screening for differentially expressed genes (DEGs)

To identify DEGs across datasets, the limma R package was utilized. DEGs were selected using the thresholds of |log2 fold change| ≥ 1 and p-value < 0.05. Volcano plots were visualized using the limma and ggplot2 R packages. A total of 33 upregulated genes were identified through Venn diagram analysis, which were subsequently considered as candidate genes for the development of a model predicting radiotherapy resistance. The Human Protein Atlas [HPA] was employed to categorize gene functions and to enrich the related pathways of the DEGs.

### 2.4 Cox regression analysis and construction of prognostic prediction model

To enhance the transparency, reproducibility, and interpretability of the prognostic model, this study adhered to the TRIPOD (Transparent Reporting of a Multivariable Prediction Model for Individual Prognosis or Diagnosis) statement [22]. The TRIPOD guideline is specifically designed to standardize the reporting of prediction model development and validation, and has been widely endorsed in biomedical research. In accordance with TRIPOD recommendations, we explicitly

reported the data sources (GEO, TCGA), inclusion criteria, definition of outcomes (treatment response and survival status), variable selection process (univariate Cox regression followed by Lasso Cox regression), model construction steps, and performance evaluation metrics (Kaplan-Meier curves, AUC, ROC analysis, and calibration curves). Additionally, clinical utility was assessed using risk stratification and subgroup comparisons. While this study is retrospective in nature and lacks external validation, we have presented complete model details and ensured transparency in all analytical procedures to facilitate reproducibility.

The predictive importance of CC-associated genes was evaluated using Cox regression analysis. The p-value, 95% CI, and hazard ratio [HR] for every gene were shown in forest plots. A gene was considered protective if its HR was less than 1, and a risk gene if its HR was more than 1. The prediction model was developed using a Lasso Cox regression model that was constructed using the glmnet R package. The genes with the highest prognostic value were selected for this model. is the formula for the proportional hazard regression model that was developed based on the findings of the univariate Cox regression. The 144 participants in the TCGA study were categorized as either high-risk or low-risk according to their median risk scores. By dividing the subjects into several groups, we were able to compare gene expression levels and conduct a survival study that linked progression-free survival [PFS] to risk ratings. The radiation resistance-related genes were tested for their predictive performance using ROC curves that were created using the survival ROC R package. The AUC value was then calculated to measure the model's ability to detect resistance. Calibration curves of nomograms were plotted to determine the accuracy of the predictions.

## 2.5 Analysis of immune infiltration

In both the high-risk and low-risk groups, Spearman's correlation was used to examine the link between gene expression levels and immune cell infiltration. Immune infiltration study, using the CIBERSORT [23] method, measured the proportions of 22 different types of immune cells in the two groups. The correlations among the immune cell types were illustrated using a correlation heatmap that was built with the "corrplot" R tool. Afterwards, we used correlation analysis to look for links to commonly found genes connected to immunological checkpoints.

Correlation analysis between risk model genes and immune cell proportions was performed using Spearman's rank correlation coefficient, as implemented in the R corrplot package. This non-parametric measure captures monotonic relationships and is widely used in immune infiltration studies. The Spearman coefficient ranges from −1 (perfect inverse relationship) to +1 (perfect direct relationship), and values exceeding this range were not observed. No non-standard correlation statistics, such as posterior Pearson or Bayesian r JZS beta, were used in this analysis.

## 2.6 Pharmacogenomic analysis

Using the Cancer Drug Sensitivity Genomics [GDSC] database, chemoresistance prediction for each tumor sample was conducted with the "oncoPredict" R package. The half-maximal inhibitory concentration [IC50] values for each chemotherapeutic agent were calculated, and regression analysis was cross-validated using the GDSC training set. Default parameters were applied, with CC cell lines selected for analysis.

## 2.7 Screening of clinical data in tissue microarrays (TMAs)

TMAs 1 and 2 comprised 31 and 13 individual 1-mm cores, respectively, representing biopsy specimens from 44 CC patients before concurrent radiotherapy. These samples were sourced from Harbin Medical University Cancer Hospital (Table 1). A TMA construction instrument was used to label representative points on paraffin specimens according to H&E-stained sections, and cores were transferred from donor blocks to recipient blocks at predetermined locations to form the arrays. The arrays were then slightly heated to fuse the cores with the recipient block. All slides required for the study were prepared by Shanghai Xinchao Biotechnology Co.

**Table 1. Clinical characteristics of samples.**

| Characteristics | TMA1 (n = 10) | TMA2 (n = 31) | Total (n = 41) |
|---|---|---|---|
| Median age (range), years | 55 (42-58) | 55(26-73) | 56 |
| Median tumor size (range), cm | 5.5(2.9-7.2) | 4.6(2.5-7.6) | 4.9 |
| FIGO stage | | | |
| I | 0 | 3 | 3 |
| II | 6 | 10 | 16 |
| III | 4 | 18 | 22 |
| Treatment response | | | |
| Resistance | 4 | 8 | 12 |
| Sensitivity | 6 | 23 | 29 |

Complete follow-up information was available for 10 out of the 13 specimens in TMA 1. Three patients lacking follow-up data were excluded. The remaining cases [n = 41) consisted of patients undergoing their first concurrent radiotherapy. Changes in tumor size were used as objective metrics to assess the efficacy of radiation therapy. After treatment, patients' reactions were assessed using RECIST 1.1, which stands for Response Evaluation Criteria in Solid Tumors [24]. Disease progression (PD) is defined as an increase of at least 20% in the total diameter of target lesions. Partial response (PR) is characterized by a reduction of ≥30% in the sum of the maximum diameters of the target lesions. If the sum of the maximum diameters decreases but fails to meet the criteria for PR, or increases but does not qualify as PD, the condition is classified as stable disease (SD). The 41 patients were divided into radiotherapy-sensitive (CR + PR) and radiotherapy-resistant (SD + PD) groups, with 12 patients classified as resistant.

In this study, all participant information was de-identified during data collection and analysis. Researchers are unable to identify individuals by any information in the data such as age, gender, clinical indicators, etc. The research received approval from the Harbin Medical University Cancer Hospital (KY2019−07) on 15/05/2019, and all experimental procedures were conducted in alignment with established ethical guidelines. And we collected information from patients who underwent concurrent chemoradiotherapy for cervical cancer from 24/12/2019–18/06/2021. All participants provided informed consent.

## 2.8 IHC

IHC was performed to evaluate the expression of genes within the TMA according to a standard protocol:

1. Deparaffinization: Tissue sections underwent deparaffinization using xylene, with the first xylene treatment lasting 15–20 minutes, followed by a second treatment of the same duration. This was succeeded by a rehydration process through a series of graded ethanol solutions: anhydrous ethanol I for 5 minutes, anhydrous ethanol II for 5 minutes, then 95% ethanol for 5 minutes, and finally 85% ethanol for 5 minutes, before transitioning to water.

2. Antigen Retrieval: Sections were incubated in a repair box containing EDTA (pH 9.0) and heated in a microwave (medium heat for 8 minutes, rested for 8 minutes, followed by medium-low heat for 7 minutes), ensuring minimal evaporation.

3. Blocking of Endogenous Peroxidase: Sections underwent incubation with 3% hydrogen peroxide for a duration of 25 minutes at room temperature, shielded from light exposure, followed by three washes in PBS, each lasting 5 minutes.

4. Serum Blocking: Sections were incubated with 3% BSA for 30 minutes at room temperature in a humidified chamber.

5. Primary Antibody Incubation: After removing excess BSA, the primary antibody was applied dropwise to the tissue and incubated overnight at 4°C in a humidified chamber.

6. Secondary Antibody Incubation: Following the acclimatization of the sections to room temperature, they underwent three washes in PBS, each lasting 5 minutes, before being incubated with the secondary antibody for a duration of 1 hour at room temperature. The sections underwent an additional washing step in PBS, repeated three times for a duration of 5 minutes each.

7. DAB Chromatography: After air drying, DAB solution was applied dropwise and incubated until the desired stain intensity was reached, monitored under a microscope. The reaction was stopped by rinsing in tap water.

8. Hematoxylin Counterstaining and digital imaging were performed. Images were analyzed using ImageJ software (Fiji, version 2.12.0).

IHC Mean Optical Density (MOD) represents the average optical density calculated by image processing software for antibody staining in tissue sections or cell cultures. TMA spots were imaged using appropriate magnification, and Image-Pro Plus 6.0 software was used to measure the integrated optical density (IOD) and positive pixel area for each image, setting a uniform standard of brownish-yellow for positive staining. MOD was calculated as: MOD = Positive IOD/ Positive pixel area.

## 2.9 Statistical analysis

Statistical analyses were conducted utilizing R software (version 4.4.1) alongside SPSS Statistics. The t-test was utilized to examine the differences between two groups, while overall survival (OS) variations were evaluated using Kaplan-Meier survival analysis, subsequently analyzed with the log-rank test. The diagnostic significance of the four target genes and the recognized risk groups for CC was assessed using receiver operating characteristic (ROC) curves. The Spearman's correlation test was employed to investigate the association between gene expression levels and the infiltration of immune cells. A p-value below 0.05 was deemed to indicate statistical significance. All analyses were performed utilizing Graph-Pad Prism (version 10.1.2).

## 3 Results

### 3.1 Transcriptome sequencing analysis of differential expression before and after Concurrent Radiotherapy for CC

To assess the impact of radiotherapy on key genes in CC, we utilized the GSE236738 dataset. This dataset comprises 31,008 single-cell RNA-sequencing data points derived from tumor samples collected from three patients prior to treatment (TN) and after CCRT. The analysis of single-cell RNA sequencing (scRNA-seq) was conducted utilizing the Seurat package. Unsupervised clustering and uniform manifold approximation and projection (UMAP) were utilized on the dataset, comprising 15,403 cells from TN samples and 15,605 cells from CCRT-treated samples (Fig 1A). The identified immune cell clusters were annotated using differential gene expression analysis(Fig 1B), utilizing markers for endothelial cells (PECAM1, VWF), mast cells (TPSB2, CPA3), fibroblasts (COL3A1, LUM), B-cells (CD79A, CD79B), macrophages (AIF1, CD68), and T cells (CD2, CD3E, TRAC) (Fig 1C and 1D).

### 3.2 Screening of key radiotherapy resistance genes associated with clinical outcomes in CC

To explore genes that contribute to radiotherapy resistance, dataset GSE56363 was selected. The examination of the dataset uncovered 275 genes exhibiting upregulation and 578 genes demonstrating downregulation subsequent to radiation exposure, specifically in patients who underwent a minimum of 46 Gy of external beam radiation therapy (EBRT) and at least 10 Gy of brachytherapy (BT). The assessment of differential gene expression was conducted utilizing the established cut-off criteria of |Log2-fold change| ≥ 1 and a p-value threshold of < 0.05 (Fig 1E). Analyses of the upregulated genes through Gene Ontology (GO) and pathway assessments, utilizing the clusterProfiler R package, revealed notable enrichment in processes associated with chromosome segregation, nuclear division, nucleochromosome

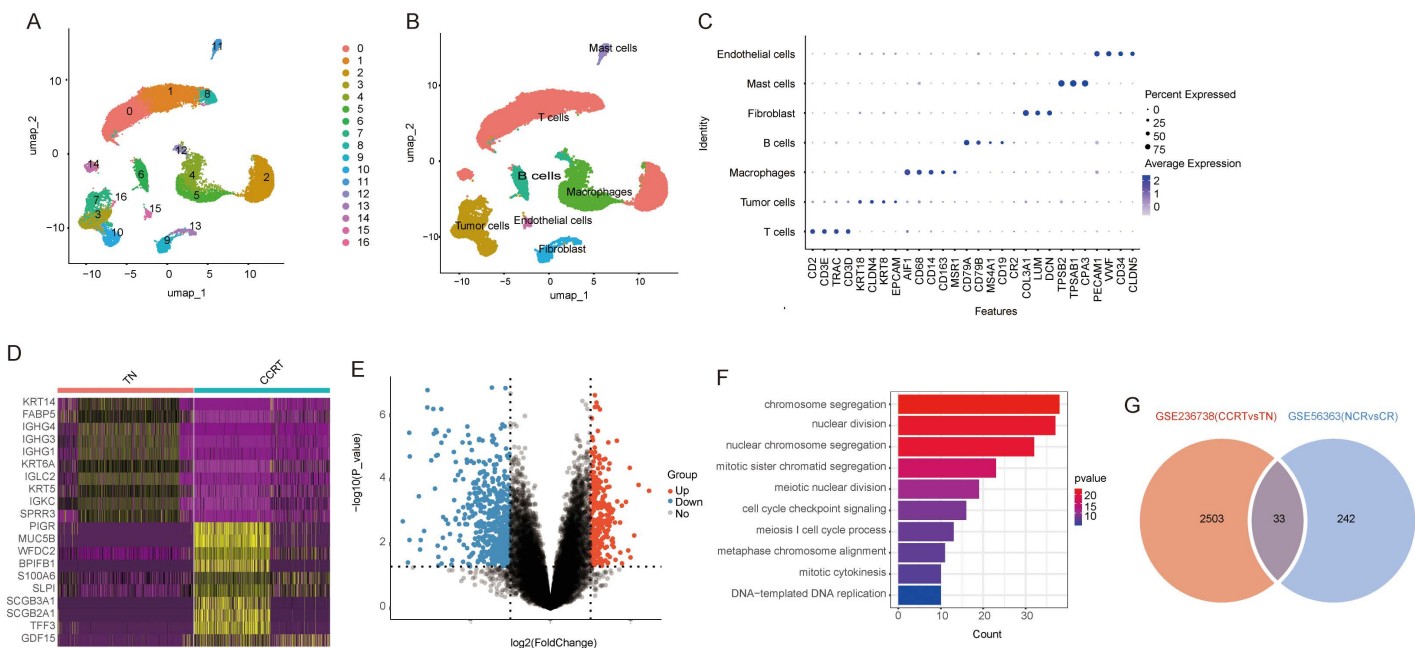

**Fig 1. Analysis of scRNA-seq and bulk RNA sequencing data. (A)** UMAP plots for TN and CCRT groups; **(B)** UMAP embedding of single-nucleus profiles from TN and CCRT samples, with cell types color-coded; **(C)** Bubble charts illustrating major cell type marker genes for each cell cluster; **(D)** Heatmap depicting differentially expressed genes (DEGs) between TN (red) and CCRT (blue) in CC, with purple indicating low expression and yellow indicating high expression; **(E)** Volcano plot of DEGs, with red representing upregulated genes and blue downregulated genes; **(F)** Pathway enrichment analysis of 275 upregulated genes; **(G)** Venn diagram illustrating the overlap of upregulated genes from GSE236738 and GSE56363, identifying 33 co-upregulated genes.

segregation, and mitotic sister chromatid segregation (Fig 1F). These results align with previous findings that focal chromosome amplification contributes to cancer development by promoting oncogene overexpression and resistance to cancer treatments through increased expression of genes that reduce the efficacy of anticancer therapies [25,26]. Additionally, DEGs were analyzed in dataset GSE236738 for pre-treatment and post-radiotherapy treated tissues, revealing 2,503 up-regulated genes. By comparing these up-regulated genes with those from dataset GSE56363 using Venn diagrams, 33 co-upregulated genes were identified and considered as candidates for constructing a model of radiotherapy resistance genes (Fig 1G).

### 3.3 Screening of target genes associated with resistance to cervical radiotherapy and construction and validation of prognostic risk models

To enhance the validation of critical genes that impact prognosis in cervical cancer patients receiving radiotherapy, we identified a cohort of 144 patients from the TCGA database for the construction of both univariate and multivariate Cox proportional hazards models. Among the 33 co-expressed upregulated genes previously identified, six genes were found to be associated with progression-free survival in the univariate Cox regression model. An HR exceeding 1 suggests that elevated expression of these genes correlates with a heightened probability of tumor progression after radiotherapy (Fig 2A). To enhance the predictive significance of these genes, Lasso-Cox regression models were developed utilizing the "glmnet" R package. According to the minimum penalty parameter ($\lambda$), four genes were preserved: MPP5, SNX7, LSM12, and GALNT3 (Fig 2B and 2C). A prognostic risk score formula was subsequently developed through multivariate Cox regression analysis (Fig 2D). The median risk score for the 144 patients was determined using this formula, leading to the

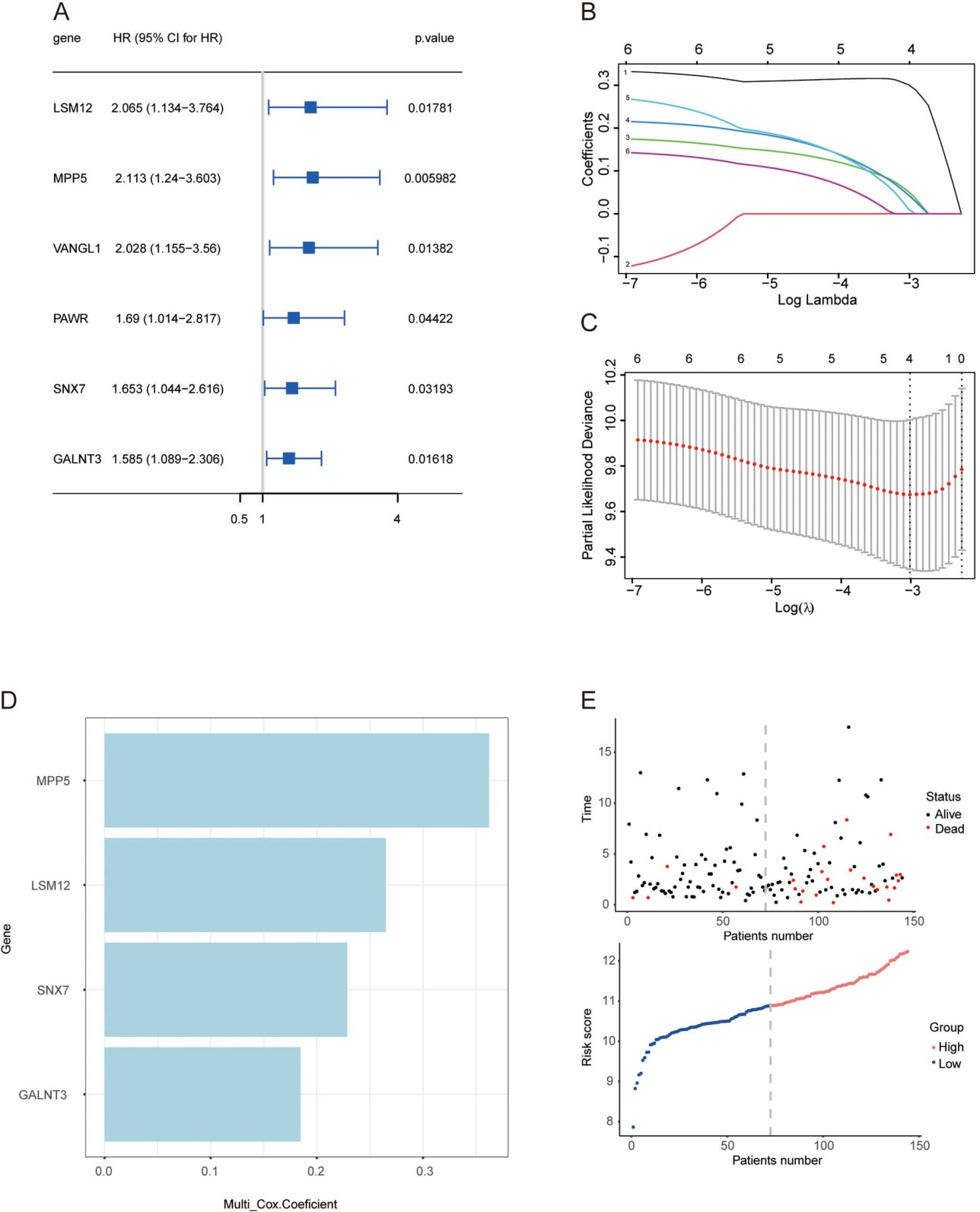

**Fig 2. Construction of a radiation-resistance gene panel in CC patients to predict prognosis. (A)** Forest plot of six candidate genes predictive of radiotherapy outcomes; **(B)** LASSO coefficient curves for the six gene markers; **(C)** Partial likelihood deviance distribution for LASSO coefficients; **(D)** Construction of a risk score based on the Cox model; **(E)** Risk score distribution and PFS of enrolled patients.

classification of patients into high-risk and low-risk groups according to this median value. Fig 2E illustrates the distribution of risk scores alongside the associated survival status.

A heatmap was created to depict the expression levels of the four target genes in individual patients (Fig 3A). Further examination of the TCGA dataset revealed markedly elevated expression levels of these genes within the high-risk cohort (Fig 3B). Furthermore, an elevated risk score correlated with a heightened risk of mortality and reduced survival durations. The Kaplan-Meier survival analysis indicated that the low-risk group exhibited a markedly improved survival outcome (P = 0.00038, Fig 3C). However, PFS curves based on individual key gene expressions showed significant results for only two genes (LSM12, P = 0.0108; MPP5, P = 0.028196. S1 Fig).

The prognostic risk model's predictive capability was demonstrated through ROC curves, with AUC values for the 1-, 2-, 3-, and 5-year predictions being 0.51, 0.64, 0.75, and 0.66, respectively, underscoring the model's reliability (Fig 3D). Calibration curves for the 1-, 2-, 3-, and 5-year overall survival probabilities confirmed good agreement between the predicted and observed outcomes (Fig 3E).

However, the relatively low AUC at 1 year (0.51) suggests that the model has limited predictive power for short-term outcomes. This may be attributed to the complexity and heterogeneity of early treatment responses, which might not be fully captured by baseline gene expression profiles. In contrast, the model demonstrated moderate to good discriminatory performance in the medium- to long-term follow-up (e.g., AUC = 0.75 at 3 years), indicating its greater utility in longer-term

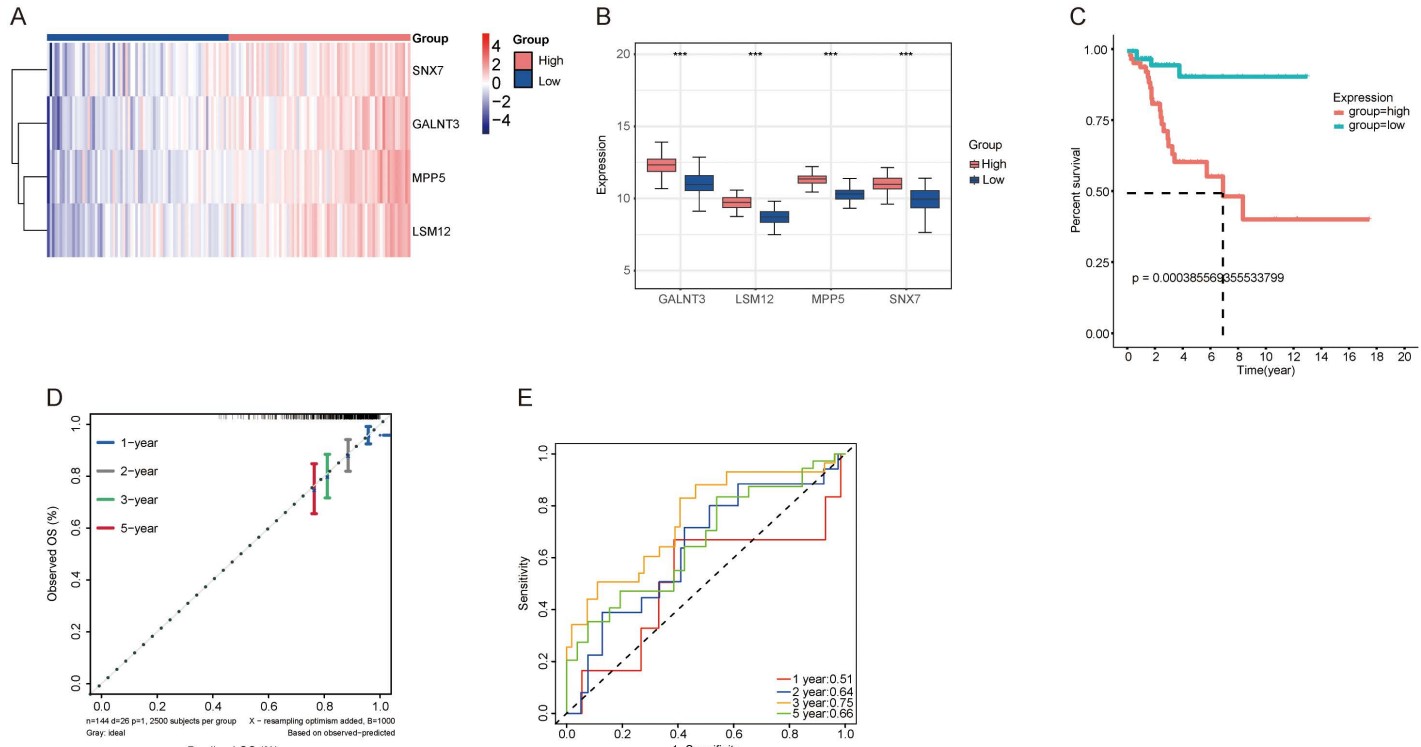

**Fig 3. Cervical cancer survival analysis and target genes expression. (A)** A heatmap depicting the expression levels of four genes across 144 patients categorized into two risk groups based on the median score. The color gradient ranges from red (high expression) to blue (low expression); **(B)** The expression levels of GALNT3, LSM1, MPP5, and SNX7 were significantly elevated in the high-risk group compared to the low-risk group of CC patients; **(C)** Survival curves for both groups, based on PFS and follow-up duration; **(D)** A ROC curve demonstrating the prognostic accuracy of the model over one, two, three, and five years; **(E)** Calibration curves of a nomogram predicting overall survival (OS) at 1, 2, 3, and 5 years, based on data from the TCGA dataset.

prognostic assessment. Therefore, while the overall performance supports its clinical potential, the model should be interpreted with caution when applied to short-term outcome prediction.

### 3.4 Differential expression of infiltrating immune cells between high and low risk groups

In the heatmap (Fig 4A), the representation of immune cells is based on their normalized relative abundance. The examination indicated that within the infiltrating immune cell population, naïve B cells, activated dendritic cells, resting memory CD4 + T cells, and regulatory T cells (Tregs) were the most notable constituents. The relationship among 22 immune cells in CC tissues was evaluated, revealing that the majority exhibited a robust positive correlation with immunity scores (Fig 4B). For instance, eosinophils exhibited a positive correlation with macrophages and activated mast cells, whereas resting mast cells demonstrated a positive correlation with activated dendritic cells and activated NK cells. Tregs showed positive relationships with activated NK cells and monocytes, while CD8 + T cells displayed negative correlations with macrophages and resting memory CD4 + T cells. To clarify the intrinsic differences in immune cell infiltration that lead to prognostic differences, differential analyses of various immune cell subtypes were conducted. The immune cells linked to the high-risk group (Fig 4C). These four target genes and risk scores were expressed at high levels in the identified immune cells (Fig 4D), aligning with the previous findings. Additionally, immune checkpoint (IC) expression profiles were examined in 144 patient samples from TCGA, showing elevated IC levels in the high-risk group (Fig 4E). Moreover, significant differences in IC expression levels were observed between the high-risk and low-risk groups (Fig 4F).

### 3.5 Correlation analysis of chemotherapeutic agents with prognostic risk models

Adjuvant and neoadjuvant chemotherapies supplement radiotherapy for CC. Predictive algorithms estimated IC50 values to assess chemosensitivity across several agents, comparing these between high and low-risk groups. The efficacy of chemotherapy is generally attributed to its ability to halt tumor cell division and promote cell death by disrupting DNA replication, cellular metabolism, or microtubule assembly. A significant negative correlation was observed between risk scores and gemcitabine concentration (Pearson coefficient: r = −0.41, p = 0.033) [27]. Similarly, risk scores negatively correlated with the dosages of Talazoparib [28], Pevonedistat [29], AGI-5198 [30], Savolitinib [31], and Sepantronium bromide [32] (Fig 5A–E). Drug concentration estimates for these agents were markedly lower in the high-risk group, indicating a potential greater sensitivity in these patients (Fig 5G–L).

### 3.6 IHC experimental validation

To enhance the understanding of the four targeted genes' contributions to predicting radiotherapy efficacy in cervical cancer, a collection of 41 human cervical cancer tissue samples was procured from the Department of Pathology at the Third Affiliated Hospital of Harbin Medical University. The expression levels of proteins MPP5, SNX7, LSM12, and GALNT3 were assessed using immunohistochemistry. The analysis of the MOD revealed that protein expression levels were markedly elevated in the radiotherapy-resistant group (Fig 6A and 6B), reinforcing the hypothesis that these four genes may serve as effective prognostic indicators for forecasting radiotherapy outcomes in CC.

## 4 Discussion

CC ranks among the most prevalent malignant tumors in women. Treatment modalities are primarily determined by the clinical FIGO staging, incorporating surgery and concurrent radiotherapy. Additionally, advancements in immunotherapy, which involve the administration of specific cytokines, antibodies blocking negative Tregs (immune checkpoint inhibitors), engineered cell therapies [33], and lysosomal viruses [34], are revolutionizing cancer treatment. To date, immunotherapy has demonstrated promising efficacy across a broad spectrum of tumors, with common agents including PD-1 inhibitors, PD-L1 inhibitors, and CTLA-4 inhibitors (13).

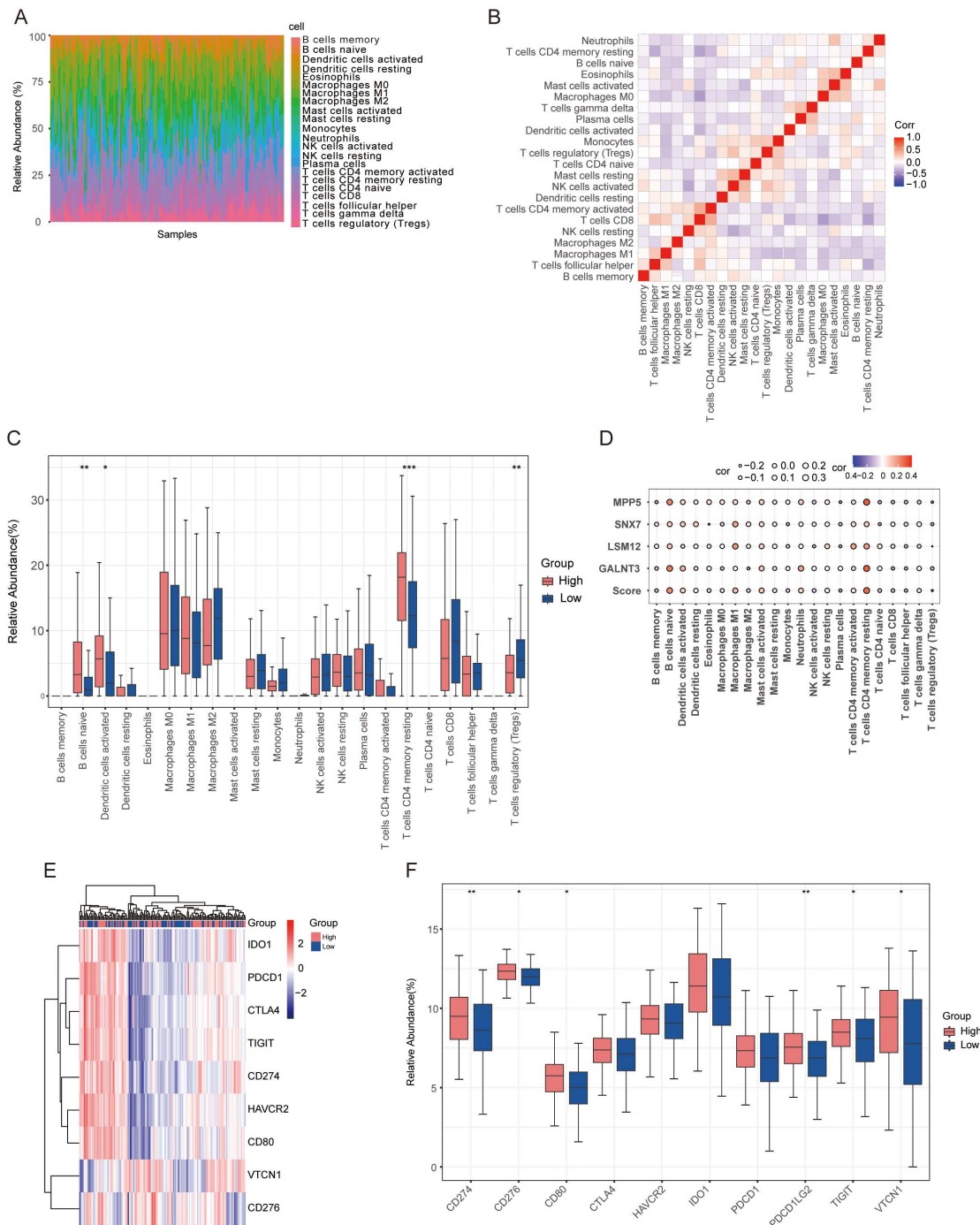

**Fig 4. Expression of IC-related genes and infiltration of immune cells in high-risk and low-risk groups. (A)** The relative proportions of 22 immune cell types in the TCGA-CC dataset; **(B)** A correlation matrix for all 22 immune cell subtypes, with correlation strengths indicated by red (high), blue (low), and white (no correlation); **(C)** Boxplots illustrating the differences in immune cell distributions between the high-risk and low-risk groups; **(D)** A bubble plot highlighting the relationships between the four genes (GALNT3, LSM1, MPP5, and SNX7) and tumor-infiltrating immune cells; **(E)** Correlations between checkpoint gene expression in the high-risk and low-risk groups; **(F)** A boxplot comparing the expression levels of checkpoint genes between the high-risk and low-risk groups.

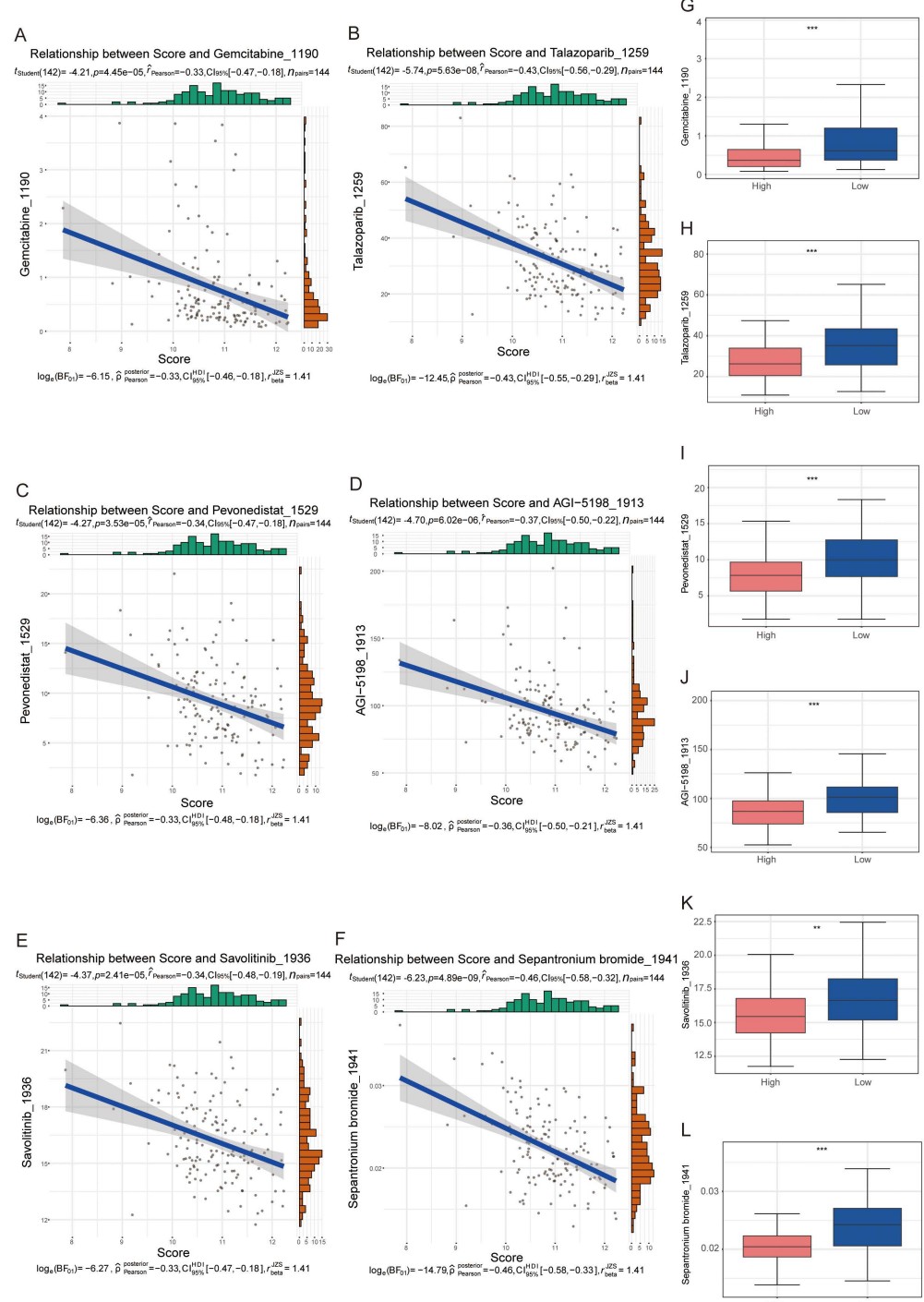

**Fig 5. IC50 of chemotherapeutic agents and Spearman correlation analysis in high-risk and low-risk groups. (A-F)** Scatter plots depicting the negative correlation between risk scores and drug IC50 (Gemcitabine, Talazoparib, Pevonedistat, AGI, Savolitinib, and Sepantronium bromide); **(G-L)** Box plots showing significantly higher sensitivity in the LSG group.

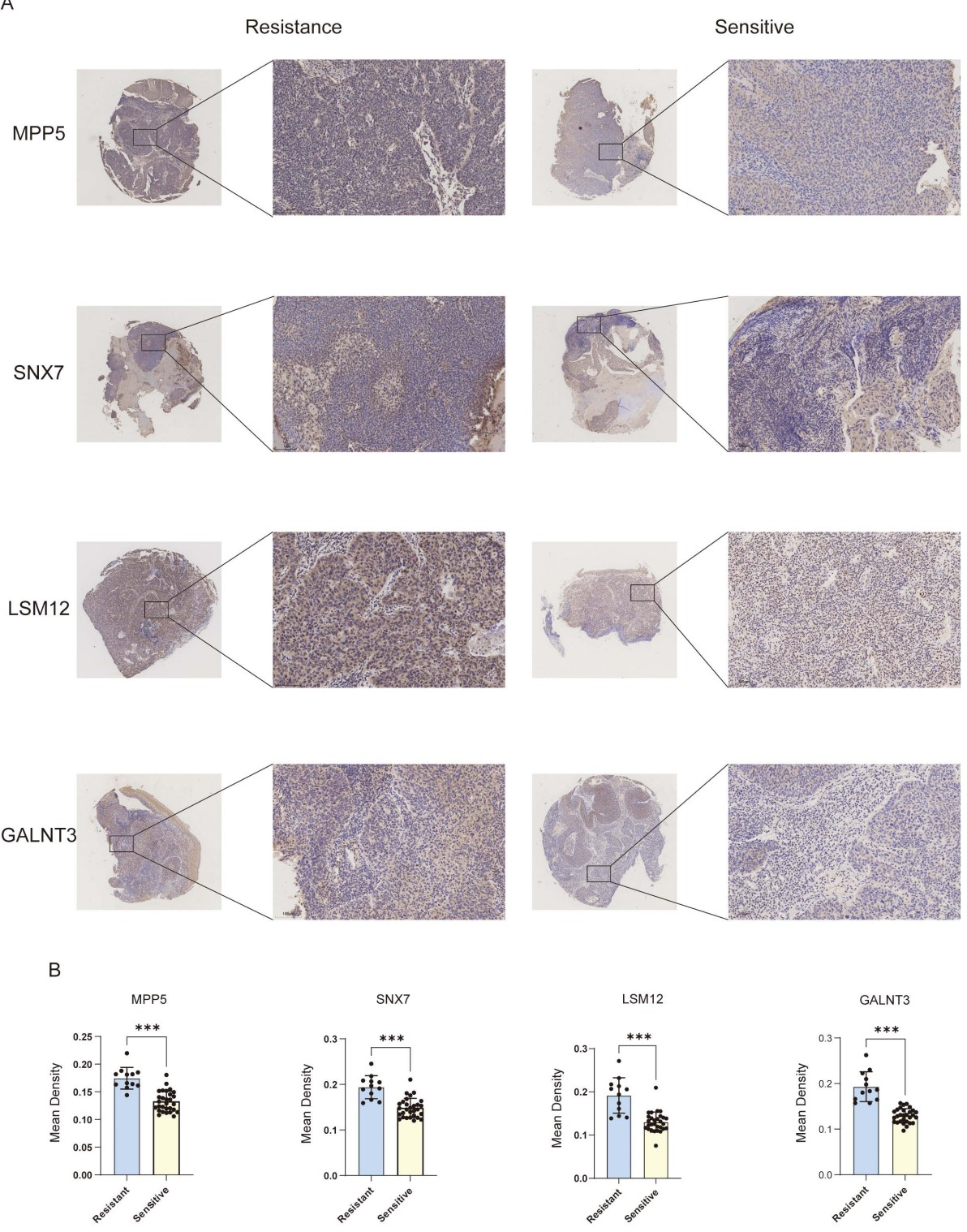

**Fig 6. IHC results for tumor tissues of patients resistant and sensitive to radiotherapy. (A)** IHC staining detected expression levels of four key genes (MPP5, SNX7, LSM12, and GALNT3) in CC tissues; **(B)** Quantification of four key radiation resistance genes in randomly selected fields of view.

Concurrent radiotherapy is recommended for all clinical stages of CC. In early-stage disease (e.g., stages IA to IB2), radiotherapy's efficacy parallels that of surgery, exhibiting comparable five-year survival, mortality, and complication rates. For advanced stages (e.g., stage IIb and beyond), radiotherapy remains the cornerstone of therapy, achieving five-year survival rates between 30% and 50%. Radiotherapy is believed to confer therapeutic benefits predominantly through inducing irreparable DNA damage and halting further cell replication and division. Recent studies, however, have highlighted its potential to trigger various immunostimulatory effects, including the release of tumor-associated antigens (TAAs) and damage-associated molecular patterns (DAMPs), which collectively enhance immune cell activation and disrupt tumor-supporting immunosuppressive environments [35].

Recent research advances have demonstrated significant efficacy in the combination of immunotherapy and radiotherapy, as well as induction chemotherapy with radiotherapy. These studies not only expand therapeutic options for patients with locally advanced CC but also lay a crucial foundation for the optimization of future treatment strategies. Further studies are required to explore biomarker screening to accurately identify patient groups that would benefit from combination therapy.

The prognostic risk model constructed in this study, based on the four genes MPP5, SNX7, LSM12, and GALNT3, demonstrated promising predictive performance for radiotherapy outcomes in cervical cancer patients, particularly in the medium- to long-term follow-up, with an AUC of 0.75 at 3 years. However, its short-term predictive ability was relatively limited, as reflected by a lower AUC of 0.51 at 1 year. This finding may indicate that early clinical outcomes following radiotherapy are influenced by a range of complex factors, including substantial heterogeneity in initial treatment responses, which may not be fully captured by baseline gene expression profiles.

Moreover, the variability in clinical data and the limited sample size may also have contributed to the reduced stability of the model's short-term predictions. These factors highlight the challenges in developing robust short-term prognostic models and underscore the need for incorporating additional clinical indicators and real-time monitoring data in future studies to enhance predictive accuracy during the early stages of treatment.

In this study, datasets related to radiotherapy were utilized, and experimental validation was performed. A total of 33 up-regulated genes were identified by integrating and differentially analyzing single-cell sequencing data from CC patients undergoing radiotherapy, before and after treatment, as well as normal cervical region (NCR) tissue data from the GEO database. These genes indicated enhanced resistance to radiotherapy and increased tumor proliferation. Significant enrichment in the pathways of chromosome segregation, karyogamy, and nuclear chromosome segregation in radiotherapy-resistant cells was observed through Gene Ontology (GO) analysis, suggesting that resistant cells promote chromosome segregation more effectively than sensitive cells. Recent studies have linked mitotic prolongation with chromosomal presence to the proliferation of cancer cells harboring mutations [36]. Subsequently, a Lasso-Cox regression analysis was conducted to develop a predictive gene model, identifying four prognostic genes: MPP5, SNX7, LSM12, and GALNT3. This study suggests that radiation-resistant cells may play a crucial role in tumor regeneration, as evidenced by clinical and mRNA data from CC patients undergoing concurrent radiotherapy, sourced from the TCGA database. Kaplan-Meier survival analyses further validated the findings. The results demonstrate that the proposed risk-prognostic model exhibits strong predictive ability for CC prognosis.

MPP5, a member of the p55 subfamily of membrane-associated guanylate kinases (MAGUKs), is essential in the assembly of protein complexes at cell-cell junctions within the plasma membrane. It is specifically localized to the apical membrane domain of outer limiting membrane (OLM) junctions [37]. Recent research has highlighted MPP5's role in establishing mammalian cell polarity, crucial for tissue organization and the regulation of cell growth and division, thus impacting the structure and function of epithelial tissues. MPP5 is vital for nervous system development and bladder cancer progression [38]. It may also play an inhibitory role in hepatocarcinogenesis by regulating the Hippo/YAP signaling pathway, which is central to cell proliferation, differentiation, and apoptosis [39,40]. However, research on MPP5 remains limited, and its role in tumor progression warrants further investigation.

Sorting nexin 7 (SNX7), found in early endosomes and multivesicular bodies, plays a crucial role as a significant member of the sorting nexin family. It serves an essential function in intracellular mechanisms including phagocytosis, protein sorting, and endosomal signaling [41–43]. Furthermore, SNX7 is linked to a range of cellular processes, such as the regulation of the cell cycle, the phenomenon of cellular senescence, cell adhesion mechanisms, and the intricate processes of DNA replication and repair. The overexpression of SNX7 has been demonstrated to decrease the production of amyloid β peptide (Aβ) by promoting the lysosomal degradation of amyloid precursor protein (APP) in HEK293T cells. Recent studies indicate that the atypical expression of SNX7 may hold clinical significance in forecasting the advancement of lung cancer. For example, a predictive model that integrates SNX7 with additional genes associated with epithelial-to-mesenchymal transition (including AGL, ECM1, ENPP1, SNX7, and TSPAN12) has been established for the early prediction of lung adenocarcinoma recurrence, indicating its possible clinical relevance in the prognosis of lung cancer [44,45].

LSM12 belongs to the LSM protein family, which comprises smaller proteins (Lsm1–10) characterized by a single LSM domain, as well as larger proteins (Lsm11, 12, 14A, 14B, and 16) that feature an additional C-terminal non-LSM structural domain. While the roles of the larger LSM proteins remain somewhat elusive, the smaller LSM proteins (such as Lsm1–8) typically function as scaffolds or chaperones. They bind RNA oligonucleotides to facilitate RNA assembly, modification, transport, and degradation, thereby influencing cellular gene expression [46,47]. LSM12 plays a crucial role in the regulation of mRNA degradation and the cellular response to DNA replication stress, significantly contributing to the mechanisms underlying oxidative stress-induced DNA damage. A recent investigation revealed that LSM12 exhibits elevated expression levels in colorectal cancer tissues when compared to adjacent normal tissues. This protein operates as a nucleoplasmic transporter, playing a crucial role in sustaining the RAN gradient between the nucleus and cytoplasm [48,49]. Furthermore, findings from the GEPIA database reveal that LSM12 exhibits significant upregulation in various human malignancies [50], such as breast invasive carcinoma (BRCA) and colon adenocarcinoma (COAD), indicating its possible role in the pathogenesis of these cancer.

GALNT3, part of the GalNAc transferase (GalNAc-Ts) family, is associated with EMT and variably expressed across different tumor types, implicating it in cancer development [51]. The GALNT family, responsible for initiating O-linked glycosylation by transferring GalNAc to serine/threonine residues of receptor proteins, plays a crucial role in post-translational modifications [52,53]. GALNT3 is notably overexpressed in high-grade serous epithelial ovarian cancer, influencing metabolic pathways and post-translational modifications within ovarian cancer cells [54]. The overexpression of this factor is associated with unfavorable prognoses in advanced ovarian cancer cases. GALNT3 expression is linked to gene mutations, DNA methylation, cellular immune infiltration, and diverse immune subtypes. Furthermore, the association with immune cell infiltration and unfavorable prognosis in LUAD highlights its potential as a valuable biomarker for the early detection of LUAD [55].

The integration of bioinformatics analysis with public databases to identify potential diagnostic and prognostic biomarkers in tumors has gained significant traction. In recent years, there has been an increasing emphasis on the tumor microenvironment, especially regarding the immune microenvironment. The presence of immune cells within the tumor microenvironment indicates the immune status and is crucial in assessing the efficacy of tumor immunotherapy, as well as impacting patient outcomes. The results indicate that the four target genes identified in this research could play a role in the recruitment and regulation of immune cells, which may influence the progression of CC. The increased expression of immune checkpoints, which aids in tumor cell evasion and enhances tumor progression, showed a positive correlation with gene expression in the high-risk group. Interestingly, there was a significant upregulation of immune checkpoint molecules within this group, suggesting that these four target genes could play a role in regulating immune checkpoints and subsequently affect the prognosis of CC. In vitro experiments demonstrated that the expression rate of these genes was elevated in the radiotherapy-resistant subgroup of squamous CC. It is hypothesized that among CC patients undergoing radiotherapy, those with high scores on this prognostic risk model may enhance their sensitivity to radiotherapy with appropriate immunotherapy or targeted drug therapy, thereby improving outcomes. Further studies are needed to verify these hypotheses.

The present study still has some limitations that need to be addressed. First, the mechanisms involved in the target genes need to be elucidated. Mechanistic studies may aid in translating these key genes into therapeutic targets. Second, most clinical samples included were squamous carcinomas, with a few poorly differentiated carcinomas, limiting the pathological types available for study. To enhance the robustness of the findings, future research should include a larger cohort of clinically well-characterized patients to assess the prognostic value of gene combinations in predicting outcomes for CC patients undergoing radiotherapy. Such clinical studies would offer a more nuanced understanding of how genetic test combinations can guide therapeutic decisions. Additionally, the use of sequencing technology on patient tumor biopsies could provide insights into the mechanisms underlying radiotherapy resistance and tumor repopulation.

## 5 Conclusions

This study revealed MPP5, SNX7, LSM12, and GALNT3 as genes linked to the prognosis of cervical cancer patients receiving concurrent radiotherapy, through the analysis of the GEO database. A prognostic risk model was developed utilizing these genes. In order to assess the model's predictive capabilities, an initial validation was performed involving 144 CC patients sourced from the TCGA database, which affirmed that the model exhibited robust prognostic predictive potential. To enhance the clinical significance of the target genes, a selection of 41 CC tissue specimens was made for immunohistochemical analysis. The findings indicated that the expression levels of MPP5, SNX7, LSM12, and GALNT3 were markedly elevated in the high-risk cohort, underscoring the essential function of these genes in forecasting radiotherapy responses CC. In summary, the model can better determine the prognostic risk of patients and develop more personalized treatment plans, such as combined immunotherapy or optimized chemotherapeutic drug selection, thus improving treatment outcomes and patient survival.

The prognostic model shows greater applicability in medium- to long-term survival prediction, rather than short-term outcome estimation. Calibration curves demonstrate good predictive consistency across all time points, supporting the model's potential for clinical application. Future studies should incorporate additional clinical variables and dynamic monitoring data to enhance the accuracy of early radiotherapy response prediction.

## Supporting information

**S1 Data. TCGA gene expression profiles and clinical data.** Gene expression profiles and corresponding clinical data of TCGA patients were obtained from https://portal.gdc.cancer.gov/. Original dataset GSE236738. The original data supporting the findings of this study can be accessed at https://www.ncbi.nlm.nih.gov/geo/query/acc.cgi?acc=GSE236738. Additional GEO dataset. Additional raw data are available at https://www.ncbi.nlm.nih.gov/geo/query/acc.cgi. (ZIP)

**S1 Fig. The survival curve of the four target genes.** (A) GALNT3. (B) LSM12. (C) MPP5. (D)SNX7. (*p < 0.05, **p < 0.01, ***p < 0.001, ****p < 0.0001). (TIF)

## Acknowledgments

We express our gratitude for the provision of data by databases such as TCGA and GEO. Sincere appreciation is extended to the reviewers and editors for their valuable comments.

## Author contributions

**Conceptualization:** Siqi Yang, Qiuyue Su, Shanshan yang.

**Data curation:** Siqi Yang, Liting Liu, Jingqi Xia, Yajuan Sun.

**Formal analysis:** Siqi Yang, Qiuyue Su, Jingqi Xia, Yajuan Sun.

**Funding acquisition:** Siqi Yang, Shanshan yang.

**Investigation:** Siqi Yang.

**Methodology:** Siqi Yang, Xinyao Zhao, Shanshan yang.

**Project administration:** Siqi Yang, Shanshan yang.

**Resources:** Siqi Yang, Shanshan yang.

**Software:** Siqi Yang, Jianan Wang.

**Supervision:** Siqi Yang, Liting Liu, Jianan Wang, Shanshan yang.

**Validation:** Siqi Yang, Liting Liu, Shanshan yang.

**Visualization:** Siqi Yang.

**Writing – original draft:** Qiuyue Su, Jianan Wang, Xinyao Zhao.

**Writing – review & editing:** Jingqi Xia, Shanshan yang.

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
