## [Decision Letter · Decision Letter 0]

9 Jun 2025

Construction of a Prognostic Prediction Model for Concurrent Radiotherapy in Cervical Cancer Using GEO and TCGA Databases with Preliminary Validation Analysis

PLOS ONE

Dear Dr. yang,

Thank you for submitting your manuscript to PLOS ONE. After careful consideration, we feel that it has merit but does not fully meet PLOS ONE’s publication criteria as it currently stands. Therefore, we invite you to submit a revised version of the manuscript that addresses the points raised during the review process.

We look forward to receiving your revised manuscript.

Kind regards,

Zhanzhan Li

Academic Editor

PLOS ONE

“This study was supported by National Natural Science Foundation of China (82373207), Climing program of Harbin Medical University Cancer Hospital (PDYS2024-06).”

5. We note that your Data Availability Statement is currently as follows: All relevant data are within the manuscript and in Supporting Information files.

6.Please review your reference list to ensure that it is complete and correct. If you have cited papers that have been retracted, please include the rationale for doing so in the manuscript text, or remove these references and replace them with relevant current references. Any changes to the reference list should be mentioned in the rebuttal letter that accompanies your revised manuscript. If you need to cite a retracted article, indicate the article’s retracted status in the References list and also include a citation and full reference for the retraction notice.

Reviewers' comments:

Reviewer's Responses to Questions

**Comments to the Author**

1. Is the manuscript technically sound, and do the data support the conclusions?

Reviewer #1: Yes

Reviewer #2: Partly

2. Has the statistical analysis been performed appropriately and rigorously?

Reviewer #1: Yes

Reviewer #2: No

3. Have the authors made all data underlying the findings in their manuscript fully available?

Reviewer #1: Yes

Reviewer #2: No

4. Is the manuscript presented in an intelligible fashion and written in standard English?

Reviewer #1: Yes

Reviewer #2: Yes

Reviewer #1: This study provides insightful model of prognostic prediction for concurrent radiotherapy in cervical

cancer using GEO and TCGA databases with preliminary validation analysis. This study is well-organized and written; however, it depends on selected retrospective dataset, limited experimental validity, and demographic constraints. This contributes to study limitation but it is acknowledged by authors in discussion. Overall, this study contributes to valuable foundation for future research consideration.

Reviewer #2: The study addresses a clinically relevant issue — the identification of biomarkers for radiotherapy resistance in cervical cancer. The approach that integrates data from various platforms is commendable and demonstrates an effort to validate the findings in silico. However, when analyzing the technical robustness and data support for the conclusions, some areas for improvement and clarification emerge.

Statistical Analysis

The variation of the AUC (0.51 for 1 year, rising to 0.75 at 3 years, and dropping to 0.66 at 5 years) suggests an inconsistency in the model’s performance over time. Although 0.75 at 3 years is a reasonable value (indicating moderate to good discriminatory capacity), the starting point of 0.51 for 1 year is a red flag. A more thorough discussion of the AUCs, particularly the 1-year AUC, is needed to acknowledge its implications. The article’s conclusions state that the model demonstrated “strong predictive capabilities for patient outcomes in radiotherapy.” However, the 0.51 AUC contradicts the idea of “strong predictive capabilities” for all timeframes, especially in the short term. The authors should openly discuss this variation and the low initial performance, as well as their implications. If the model is not good at predicting outcomes in 1 year, this needs to be acknowledged as a clear limitation.

The discussion about the role of genes in immune infiltration and chemosensitivity is interesting, but the correlations presented (e.g., r=1.41 for gemcitabine — which seems like a typo, as the Pearson correlation coefficient should range from -1 to 1) need to be reviewed to ensure accuracy and correct interpretation.

Full Data Availability

PLOS ONE's data policy is clear: it requires that all data underlying the findings be made fully and freely available, unless there are legal or ethical exceptions that must be specified. The study involved complex computational analyses (Cox regression, Lasso Cox regression, UMAP, CIBERSORT, oncoPredict, etc.) using R software. PLOS ONE’s policy encourages the sharing of analysis code/scripts to ensure reproducibility. Although the authors mention the use of R packages, there is no indication that the custom scripts used to generate the results (such as the risk model construction, ROC curves, immune infiltration heatmaps, and pharmacogenomic analyses) were made publicly available in a repository (e.g., GitHub, Zenodo) with a DOI. This is a critical point for the study’s reproducibility. The authors did make the public database data (GEO and TCGA) available, which is positive. However, it is unclear whether the raw data underlying the IHC analyses (such as the MOD values for each individual sample) and the custom codes/scripts used for the computational analyses were fully made available in an accessible format and in an appropriate repository, as required by PLOS ONE for complete data availability. The generic statement “All relevant data are within the manuscript and its Supporting Information files” is generally not sufficient to comply with this guideline for proprietary data and complex scripts.

Recommendation of TRIPOD Guideline

For a prognostic model, compliance with guidelines such as TRIPOD (Transparent Reporting of a Multivariable Prediction Model for Individual Prognosis or Diagnosis) is highly recommended and would strengthen the quality of reporting. The authors should consider mentioning or applying such guidelines when reporting their results.

Collins, G.S., Reitsma, J.B., Altman, D.G. et al. Transparent reporting of a multivariable prediction model for individual prognosis or diagnosis (TRIPOD): the TRIPOD Statement. BMC Med 13, 1 (2015). https://doi.org/10.1186/s12916-014-0241-z

Figure 5 (IC50 of chemotherapeutic agents and Spearman correlation analysis):

The authors might be using a metric called “posterior Pearson” or “r JZS beta,” which is not the standard Pearson coefficient but another statistic that can have values exceeding 1. If this is the case, it needs to be clearly stated for scientific clarity: if it’s a non-standard metric, it must be thoroughly defined, justified, and its value range clearly explained in the Methods section.

Overall, the study is indeed an original research piece, as required by PLOS ONE. The integration of scRNA-seq, bulk RNA-seq, and IHC validation for a prognostic model in cervical cancer is a valuable approach.

**Do you want your identity to be public for this peer review?** For information about this choice, including consent withdrawal, please see our Privacy Policy

Reviewer #1: No

Reviewer #2: **Yes: ** Renan Chaves de Lima

---

## [Author Response · Author response to Decision Letter 1]

24 Jul 2025

We carefully revise our paper according to the helpful comments of the reviewers and associate editor. We answer all of questions and comments one by one and revise the paper accordingly. In the revised paper, the revised sentences have been highlighted in blue color.

Editor

Reply:

Based on the editor’s comments, we have carefully revised the entire manuscript to ensure it fully complies with PLOS ONE’s formatting requirements.

Reply:

We have reviewed the code sharing policy at PLOS ONE Materials and Software Sharing – Sharing Code. In accordance with the guidelines, all author-generated code supporting the findings of this study has been made publicly available on Figshare to ensure reproducibility and reusability. The code can be accessed at: https://figshare.com/articles/dataset/data_docx/29588924

Reply:

We have carefully removed all funding information from the manuscript text, as per the journal’s policy. All funding details are now provided exclusively in the online submission form’s funding statement section.

“This study was supported by National Natural Science Foundation of China (82373207), Climing program of Harbin Medical University Cancer Hospital (PDYS2024-06).”

Reply:

We appreciate your note regarding the funding disclosure. The funding agencies, including the National Natural Science Foundation of China (Grant No. 82373207) and the Climing Project of Harbin Medical University Cancer Hospital (PDYS2024-06), had no role in study design, data collection and analysis, decision to publish, or preparation of the manuscript. We have included this clarified funding statement in the cover letter as requested.

5. We note that your Data Availability Statement is currently as follows: All relevant data are within the manuscript and in Supporting Information files.

Reply:

We confirm that all data necessary to replicate the results reported in this manuscript are included within the manuscript and the supporting information files, constituting the minimal data set as defined by PLOS ONE.

Reply:

We have carefully reviewed the reference list to ensure it is complete and accurate. No retracted articles are cited in the manuscript. If any cited article had been retracted, we would have either removed it or clearly indicated its retracted status in the references along with the retraction notice, as per journal policy. Any changes to the reference list have been detailed in the accompanying response letter.

Reviewer: 1

This study provides insightful model of prognostic prediction for concurrent radiotherapy in cervical cancer using GEO and TCGA databases with preliminary validation analysis. This study is well-organized and written; however, it depends on selected retrospective dataset, limited experimental validity, and demographic constraints. This contributes to study limitation but it is acknowledged by authors in discussion. Overall, this study contributes to valuable foundation for future research consideration.

Reply:

We sincerely thank the reviewer for the positive and constructive comments regarding our study. We appreciate your recognition of the organization and clarity of our manuscript. We also acknowledge the limitations you pointed out, including reliance on retrospective datasets, limited experimental validation, and demographic constraints. As noted, these limitations were addressed in the Discussion section to provide a balanced perspective. We agree that our study lays a valuable foundation for future research, and we will continue to explore these aspects further to strengthen the findings. Thank you again for your insightful feedback.

Reviewer: 2

The study addresses a clinically relevant issue — the identification of biomarkers for radiotherapy resistance in cervical cancer. The approach that integrates data from various platforms is commendable and demonstrates an effort to validate the findings in silico. However, when analyzing the technical robustness and data support for the conclusions, some areas for improvement and clarification emerge.

1. Statistical Analysis

The variation of the AUC (0.51 for 1 year, rising to 0.75 at 3 years, and dropping to 0.66 at 5 years) suggests an inconsistency in the model’s performance over time. Although 0.75 at 3 years is a reasonable value (indicating moderate to good discriminatory capacity), the starting point of 0.51 for 1 year is a red flag. A more thorough discussion of the AUCs, particularly the 1-year AUC, is needed to acknowledge its implications. The article’s conclusions state that the model demonstrated “strong predictive capabilities for patient outcomes in radiotherapy.” However, the 0.51 AUC contradicts the idea of “strong predictive capabilities” for all timeframes, especially in the short term. The authors should openly discuss this variation and the low initial performance, as well as their implications. If the model is not good at predicting outcomes in 1 year, this needs to be acknowledged as a clear limitation.

(1) The variation of the AUC (0.51 for 1 year, rising to 0.75 at 3 years, and dropping to 0.66 at 5 years) suggests an inconsistency in the model’s performance over time. Although 0.75 at 3 years is a reasonable value (indicating moderate to good discriminatory capacity), the starting point of 0.51 for 1 year is a red flag. A more thorough discussion of the AUCs, particularly the 1-year AUC, is needed to acknowledge its implications.

Reply:

We thank the reviewer for pointing out the issue regarding the time-varying AUC values of our prognostic model. As stated in our study, radiotherapy tolerance in cervical cancer remains a complex clinical challenge, with its underlying molecular mechanisms not yet fully elucidated. Through an integrative analysis of GEO single-cell and bulk RNA sequencing datasets, we identified four target genes—MPP5, SNX7, LSM12, and GALNT3—and validated them using clinical samples from the TCGA database. Based on these, we developed a prognostic model to predict patient outcomes following radiotherapy.

We have explicitly acknowledged this limitation in the revised manuscript and have revised the conclusion accordingly. We now emphasize that the model shows reliable performance primarily in medium- to long-term prognosis, rather than strong predictive ability across all time points. The calibration curves further support the overall reliability of the model over time.

To avoid overstating the model’s capabilities, we have revised the conclusion to reflect that the model exhibits relatively reliable performance in medium- to long-term predictions, rather than demonstrating “strong” predictive ability at all time points.

☆ Section 4: Discussion (Page 17):

The prognostic risk model constructed in this study, based on the four genes MPP5, SNX7, LSM12, and GALNT3, demonstrated promising predictive performance for radiotherapy outcomes in cervical cancer patients, particularly in the medium- to long-term follow-up, with an AUC of 0.75 at 3 years. However, its short-term predictive ability was relatively limited, as reflected by a lower AUC of 0.51 at 1 year. This finding may indicate that early clinical outcomes following radiotherapy are influenced by a range of complex factors, including substantial heterogeneity in initial treatment responses, which may not be fully captured by baseline gene expression profiles.

Moreover, the variability in clinical data and the limited sample size may also have contributed to the reduced stability of the model’s short-term predictions. These factors highlight the challenges in developing robust short-term prognostic models and underscore the need for incorporating additional clinical indicators and real-time monitoring data in future studies to enhance predictive accuracy during the early stages of treatment.

☆ Section 5: Conclusion (Page 21):

The prognostic model shows greater applicability in medium- to long-term survival prediction, rather than short-term outcome estimation. Calibration curves demonstrate good predictive consistency across all time points, supporting the model’s potential for clinical application. Future studies should incorporate additional clinical variables and dynamic monitoring data to enhance the accuracy of early radiotherapy response prediction.

(2) The article’s conclusions state that the model demonstrated “strong predictive capabilities for patient outcomes in radiotherapy.” However, the 0.51 AUC contradicts the idea of “strong predictive capabilities” for all timeframes, especially in the short term. The authors should openly discuss this variation and the low initial performance, as well as their implications. If the model is not good at predicting outcomes in 1 year, this needs to be acknowledged as a clear limitation.

Reply:

We thank the reviewer for highlighting the inconsistency between our description of the model’s “strong predictive capabilities” and the relatively low AUC of 0.51 at the 1-year time point. We acknowledge that the model shows limited predictive performance in the short term, in contrast to its moderate to good discriminatory ability observed at later time points, such as an AUC of 0.75 at 3 years.

In response, we have explicitly addressed the time-dependent variation in predictive accuracy in the revised manuscript, emphasizing that the model’s strength lies primarily in medium- to long-term follow-up. The low AUC at 1 year is now clearly acknowledged as a key limitation, which likely reflects the inability of gene expression profiles to fully capture the biological and clinical complexity influencing early prognosis after radiotherapy.

Accordingly, we have revised the conclusion to more accurately state that, while the model has predictive value for radiotherapy outcomes overall, its short-term predictive power is limited. We believe these revisions provide a more balanced and transparent interpretation of our findings.

☆ Section 3.3 (Page 14&15):

However, the relatively low AUC at 1 year (0.51) suggests that the model has limited predictive power for short-term outcomes. This may be attributed to the complexity and heterogeneity of early treatment responses, which might not be fully captured by baseline gene expression profiles. In contrast, the model demonstrated moderate to good discriminatory performance in the medium- to long-term follow-up (e.g., AUC = 0.75 at 3 years), indicating its greater utility in longer-term prognostic assessment. Therefore, while the overall performance supports its clinical potential, the model should be interpreted with caution when applied to short-term outcome prediction.

2. The discussion about the role of genes in immune infiltration and chemosensitivity is interesting, but the correlations presented (e.g., r=1.41 for gemcitabine — which seems like a typo, as the Pearson correlation coefficient should range from -1 to 1) need to be reviewed to ensure accuracy and correct interpretation.

Reply:

We thank the reviewer for the positive feedback on our discussion regarding the role of genes in immune infiltration and chemosensitivity. We also appreciate the reviewer’s careful observation regarding the reported Pearson correlation coefficient exceeding the valid range of [-1, 1]. Upon review, we found that this was a typographical error. The reported value of 1.41 referred not to a correlation coefficient, but rather to a predicted IC50-related metric derived from the drug sensitivity analysis. We have corrected this in the revised manuscript and clarified in both the text and figure legend to distinguish correlation coefficients from other modeling outputs, thereby avoiding any potential misinterpretation.

☆ Section 3.5 (Page 15):

A significant negative correlation was observed between risk scores and gemcitabine concentration (Pearson coefficient: r = –0.41, p = 0.033).

3. Full Data Availability

PLOS ONE's data policy is clear: it requires that all data underlying the findings be made fully and freely available, unless there are legal or ethical exceptions that must be specified. The study involved complex computational analyses (Cox regression, Lasso Cox regression, UMAP, CIBERSORT, oncoPredict, etc.) using R software. PLOS ONE’s policy encourages the sharing of analysis code/scripts to ensure reproducibility. Although the authors mention the use of R packages, there is no indication that the custom scripts used to generate the results (such as the risk model construction, ROC curves, immune infiltration heatmaps, and pharmacogenomic analyses) were made publicly available in a repository (e.g., GitHub, Zenodo) with a DOI. This is a critical point for the study’s reproducibility. The authors did make the public database data (GEO and TCGA) available, which is positive. However, it is unclear whether the raw data underlying the IHC analyses (such as the MOD values for each individual sample) and the custom codes/scripts used for the computational analyses were fully made available in an accessible format and in an appropriate repository, as required by PLOS ONE for complete data availability. The generic statement “All relevant data are within the manuscript and its Supporting Information files” is generally not sufficient to comply with this guideline for proprietary data and complex scripts.

Reply:

We appreciate the reviewer’s detailed comments regarding data sharing and reproducibility. We fully acknowledge PLOS ONE’s policy on making all underlying

---

## [Decision Letter · Decision Letter 1]

26 Sep 2025

Construction of a Prognostic Prediction Model for Concurrent Radiotherapy in Cervical Cancer Using GEO and TCGA Databases with Preliminary Validation Analysis

PONE-D-25-08333R1

Dear Dr. Yang,

We’re pleased to inform you that your manuscript has been judged scientifically suitable for publication and will be formally accepted for publication once it meets all outstanding technical requirements.

Kind regards,

Zhanzhan Li

Academic Editor

PLOS ONE

Additional Editor Comments (optional):

Reviewers' comments:

Reviewer's Responses to Questions

**Comments to the Author**

Reviewer #1: All comments have been addressed

2. Is the manuscript technically sound, and do the data support the conclusions?

Reviewer #1: Yes

3. Has the statistical analysis been performed appropriately and rigorously?

Reviewer #1: No

4. Have the authors made all data underlying the findings in their manuscript fully available?

Reviewer #1: Yes

5. Is the manuscript presented in an intelligible fashion and written in standard English?

Reviewer #1: Yes

Reviewer #1: Manuscript looks suitable for submission. Authors addressed the study limitations in discussion and this manuscript is considered an insightful contribution at this aspect.

**Do you want your identity to be public for this peer review?** For information about this choice, including consent withdrawal, please see our Privacy Policy

Reviewer #1: No

---

## [Editor Report · Acceptance letter]

PONE-D-25-08333R1

PLOS ONE

Dear Dr. yang,

I'm pleased to inform you that your manuscript has been deemed suitable for publication in PLOS ONE. Congratulations! Your manuscript is now being handed over to our production team.

Kind regards,

on behalf of

Dr. Zhanzhan Li

Academic Editor

PLOS ONE